# Sensing HIV Protease and Its Inhibitor Using “Helical Epitope”—Imprinted Polymers

**DOI:** 10.3390/s20123592

**Published:** 2020-06-25

**Authors:** Chien-Yu Chou, Chung-Yin Lin, Cheng-Hsin Wu, Dar-Fu Tai

**Affiliations:** 1Department of Chemistry, National Dong Hwa University, Hualien 974003, Taiwan; 610312022@gms.ndhu.edu.tw (C.-Y.C.); m9812022@gms.ndhu.edu.tw (C.-H.W.); 2Medical Imaging Research Center, Institute for Radiological Research, Chang Gung University/Chang Gung Memorial Hospital, Taoyuan 33323, Taiwan; 3Department of Nephrology and Clinical Poison Center, Chang Gung Memorial Hospital, Taoyuan 333423, Taiwan

**Keywords:** molecularly imprinted polymers, quartz crystal microbalance, HIV protease, protease inhibitor, nelfinavir, helical epitope

## Abstract

A helical epitope-peptide (lle^85^-Gly^94^) was selected from the α-helix structure of the HIV protease (PR) as the template, which represents an intricate interplay between structure conformation and dimerization. The peptide template was mixed with water, trifluoroethanol (TFE), and acetonitrile (ACN) at a certain ratio to enlarge the helical conformation in the solution for the fabrication of helical epitope-mediated molecularly imprinted polymers (HEMIPs) on a quartz crystal microbalance (QCM) chip. The template molecules were then removed under equilibrium batch rebinding conditions involving 5% acetic acid/water. The resulting HEMIPs chip exhibited a high affinity toward template peptide HIV PR_85–94_, His-tagged HIV PR, and HIV PR, with dissociation constants (K_d_) as 160, 43.3, and 78.5 pM, respectively. The detection limit of the developed HIV PR_85–94_ QCM sensor is 0.1 ng/mL. The HEMIPs chip exhibited a high affinity and selectivity to bind HIV PR and subsequently to an inhibitor of HIV PR (nelfinavir). The HIV PR binding site was properly oriented on the HEMIPs-chip to develop a HIV PR/HEMIPs chip, which can effectively bind nelfinavir to establish a sandwich assay. The nelfinavir then attached to the HIV PR/HEMIPs chip, which can be easily removed involving 0.8% acetic acid/water. Therefore, HIV PR/HEMIPs chip can be useful to screen for other HIV PR inhibitors. This technique may improve drug targeting for HIV therapy and also strengthen investigations into other virus assays.

## 1. Introduction

Molecular imprinting is a powerful technique that can directly generate artificial receptors on sensors for specific detection [1]. Imprints of template molecules are obtained via a synthetic organic polymer matrix [2,3]. After the removal of a template from a rigid network matrix, the recognition sites complementary to the template molecules can be obtained according to either their structural, geometrical, and chemical features or the position of the functional groups [3]. Therefore, molecularly imprinted polymers (MIPs) have been widely used for target analytes, such as myoglobin biomarkers and hemoglobin variants, for protein recognition and purification [4,5]. On the other hand, quartz crystal microbalance (QCM) has been demonstrated as a powerful device. The QCM chip’s gold flat surface is suitable for coating MIPs. Its quartz crystal is sensitive to mass changes and serves as a signal transducer [6,7]. The MIPs-QCM is a convenient chemical sensor to detect binding signals [8,9].

Viruses are the most dangerous threat to human health nowadays. Acquired immunodeficiency syndrome (AIDS) is one of the most severe illness caused by the human immunodeficiency virus (HIV) [10]. HIV protease (HIV PR) is a virus-specific aspartic PR responsible for the proteolytic maturation of the polyproteins of gag and gag-pol during virion maturation and for the proliferation of the retrovirus [11,12]. HIV PR is also a dimer of two 99-amino acid monomer subunits [13]. The terminal domain of each subunit is made up of β-sheets 1–4 and 95–99 in each monomer, the random coil residues 4–9, and the α-helix residues 85–94 in each monomer [14,15]. Gly^86^-Arg^87^-Asn^88^ in the α-helix is unique on the surface of retroviral PR [16], corresponding to an effective neutralizing antibody [14,15,16]. Since virion maturation is essential for viral infectivity, HIV PR is an excellent target for drug therapy because the interference with PR activity blocks the maturation of the virus, preventing both virus replication and the further infection of host cells [12]. Therefore, the rational design of a preventive PR inhibitor requires to understand the antibody–epitope interaction for antiretroviral therapies specific to HIV/AIDS [17,18].

Previously, several approaches have been developed to detect HIV PR and to screen for its inhibitors. These approaches rest on high-performance liquid chromatography (HPLC) separation or on chromophoric, fluorogenic, or radiolabeled synthetic peptides [19,20]. However, these current protocols are complex and time-consuming. In addition, identifying the structure of HIV PR is a prerequisite for analyzing the exposed surfaces of the target sites for mapping conformational epitopes [21]. Therefore, an epitope template-based system is simpler and safer for detecting HIV PR and screening its inhibitors. Our studies have shown that an MIP-coated QCM sensor using epitope imprinting significantly strengthens the detection of protective antigens and their segments [22,23]. These results also suggest that functionalized epitope cavities on a QCM surface can effectively catch an oriental antigen. It is promising to use an oriented protein as a receptor for the screening of its substrate or analogs.

The aim of the present study was to develop a highly sensitive sandwich assay through the generation of helical epitope-mediated molecularly imprinted polymers (HEMIPs) on a QCM chip, to orientation bind HIV PR for the screening of HIV PR inhibitors. To date, there is no report using a HEMIPs-QCM system for detecting HIV PR and screening its inhibitors. The assay is based on the polymerization of a family of acrylamide-based monomers and crosslinkers in the presence of a template molecule (helical peptide segments of HIV PR), resulting in coating HEMIPs on a QCM chip. As expected, a helical epitope-mediated QCM nanosensor detected template peptides and mother proteins. A HEMIPs-QCM system is able to screen for HIV PR inhibitors with a HIV PR/HEMIPs chip. This cell-free approach thus offers a safe and effective process by which helical epitope cavities can be useful to sense virus-related proteins and its inhibitors.

## 2. Materials and Methods

### 2.1. Materials and Devices

The 10-mer HIV PR_85–94_ IGRNLLTQIG was synthesized using a CEM Discover Microwave Synthesizer (Kohan Co., Taipei, Taiwan) at National Dong Hwa University. Then, (*N*-Acr-l-Cys-NHBn)_2_ was synthesized as previously described [24,25]. The buffer used for all experiments was a PBS (20 mM NaH_2_PO_4_, pH 7.4). All reagents were reagent grade unless otherwise stated. 

The QCM chip showed a reproducibility of ±0.1 Hz, involved 9-MHz, had a diameter of 4.5 mm, and featured an AT-cut quartz-crystal wafer with circular gold electrodes on both sides. The chip was obtained from Tai-Yi Electronic Co. (Taipei, Taiwan). The product compound was determined by using both an Intelligent HPLC system (Hitachi Corp., Tokyo, Japan), a Vercopak-RP C18 column (Vercotech Corp., Taipei, Taiwan), and a BrukerAutoflex MALDI/TOF mass spectrometer (Bremen, Germany). The secondary conformations of the HIV PR_85–94_ peptide, before and after interaction with various solutions, were characterized with a JASCO, J-715 (Japan) circular dichroism (CD) spectrometer in the wavelength range of 195–255 nm.

### 2.2. Preparation of a “Molecularly Imprinted Polymers”-Coated QCM Chip

The QCM chips were immersed in a 19 μmol solution of (*N*-Acr-l-Cys-NHBn)_2_ in an aqueous mixture of 10 mL of HPLC-grade acetonitrile (ACN) and 0.1 mL of dimethylformamide (DMF) for 48 h, and were then rinsed exhaustively with ACN and dried under vacuum. A solution of epitope peptide (3 μmol) and acrylic acid (55 μmol), acrylamide (55 μmol), *N*-benzylacrylamide (110 μmol), and *N*,*N*-ethylene-bis-acrylamide (220 μmol) was mixed thoroughly in 0.3 mL of an aqueous solution (trifluoroethanol (TFE)/ACN/ deionized water (DI) water). After pipetting 2 μL of the aliquot on top of the gold electrode-(*N*-Acr-l-Cys-NHBn)_2_, the chip was placed horizontally into a 20 mL glass vial. Then, the vial was irradiated with UV light at 350 nm for 6 h. A thin film on the gold surface was first washed with 5% acetic acid/water to remove the template. This resultant chip was then washed with deionized water and dried under vacuum for later use.

### 2.3. QCM Equipment

All adsorption experiments were performed by using not only a flow-injection system outfitted with an HPLC pump (model L7110, Hitachi, flow rate 0.1 mL/min), but also a home-built flow cell, a sample injection valve (model 1106, OMNIFIT), a home-built oscillation circuit (including an oscillator and a frequency counter), and a personal computer. The polymer-coated QCM chip was placed into the flow cell. PBS and 0.5% acetic acid were respectively used and injected it into the flow system to equilibrate the newly imprinted chips quickly. 

### 2.4. QCM Adsorption Measurement

All adsorption experiments were performed using QCM equipped with a flow injection analysis (FIA). For each measurement, PBS was first flowed through the cell to obtain a stable baseline. Then, the template peptide or protein with different concentrations was injected into the cell. The binding of the HIV PR epitope templates or proteins to HEMIPs film caused a mass change reflected in the oscillation frequency [26]. The oscillation frequency gradually decreased while the adsorbed analyte increased. The data obtained were plotted with the saturation equation for specific binding ($B=B_{\max}c/(K_{d}+c)$, $B=H/M_{w}$, where c is the concentration of the analytes in the solution, B is the amounts of analytes bound, $M_{w}$ is the molecular weight of the analyte, H is the frequency shifts observed in the QCM, $B_{\max}$ is set as the maximum amount of analyte bound, B is the amount of analytes bound, and K_d_ is the dissociation constant). For the determination of the binding affinity of HIV PR to HIV PR with an inhibitor, a similar approach was used. HIV PR (1 μg/mL) was bound on top of the helical epitope-mediated MIPs-QCM (HEMIPs-QCM) chips. Upon saturation, nelfinavir (HIV PR inhibitor) was injected at a concentration of 100 ng/mL until reaching equilibrium. We washed the chip with 0.5% acetic acid in DI water or washed with 0.8% acetic acid in DI water, the HIV PR inhibitor was removed from the surface of HEMIPs-HIV PR. Reuse of the HEMIPs-HIV PR chip was able to regenerate the “sandwich” layers needed to capture and detect the HIV PR inhibitor again.

## 3. Results and Discussion

### 3.1. Identification and Selection of an Epitope Template from HIV Protease (HIV PR)

HIV PR is essential to the generation of mature enzymes and structural components in the production of the HIV infectious virus. Interactions between the two subunits stabilize the interface of a free PR dimer [27,28]. Each subunit has a secondary structure consisting mostly of beta strands and involving a short α-helix. Because the α-helix is a common motif for the secondary structure of proteins, as well as of the recognition sites for other proteins [29], the α-helix hydrogen bonds observed between peptide analogs and the conserved regions of HIV PR constitute a space for the design of non-peptide inhibitors with equivalent polar interactions (Figure 1) [11]. Sequence 85–94 was indicated as the helical epitope of HIV PR [16]. This 10-mer peptide was synthesized, pooled, and lyophilized, resulting in an approximately 97% purity of white solid by HPLC analysis. Using MALDS-TOF-MS, the molecular weights (MWs) of the synthetic peptide was confirmed. The typical analysis gave an MW of 1084.241 g/mol as expected. The result indicates that the synthetic epitope peptide IGRNLLTQIG has the 85–94 sequence of HIV PR.

### 3.2. Helical Structure Analysis

It is essential to test whether or not purified HIV PR_85–94_ peptides possess conformational similarity to the helix of HIV PR for the MIP platform. The CD spectra of the HIV PR_85–94_ peptide in different ratios of trifluoroethanol (TFE), acetonitrile (ACN), and DI water revealed the existence of basic ordered structures (Figure 2A). There is a sign of possible interaction between the unordered (random coil) and ordered (α-helix or β-sheet) structures in the secondary structure of conformation changes (Figure 2B). The spectra are characterized by negative bands ranging from ~195 to ~215 nm. The signal peptide peak and its variants exhibited conformational behavior similar to that of the α-helix structure, as it was witnessed in the presence of two negative bands at 202 and 205 nm for the aqueous solution of TFE:DI = 7:3. These two peaks corresponding to the α-helical structure resulted from n–π* and π–π* transitions [29]. n–π* is responsible for the negative band at 205 nm and the negative band at 202 nm. Moreover, upon analyzing the characteristic bands of aqueous mixture solutions of TFE:DI = 7:3, TFE:DI = 3:7, or CAN:DI = 5:5, the spectra is corresponded to a combination of α-helix, β-sheet, β-turn and the random coil structures. Our results demonstrate that the selected peptide template, when dissolved at TFE: DI = 7:3, could be significant for ordered interactions or helical activity.

### 3.3. Imprinting Effect and Binding Interaction

The HIV PR_85–94_ peptide templated HEMIPs were prepared as a co-polymer on a QCM chip using acrylic acid, acrylamide, *N*-benzylamide and EBAA (*N*,*N’*-ethylene bisacrylamide), in line with our previous studies [22,23,24,25]. First, an amino acid cross-linker (*N*-Acr-l-Cys-NHBn)_2_ was used as a thiol-ene monomer, which enables simple functionalization with binding groups for the polymer’s immobilization of gold chips and an initiator for UV-irradiated photopolymerization. Polymer surfaces were stamped with a bounded 10-mer peptide template (HIV PR_85–94_). To wash out the templates, we treated the formatted chips with alkaline, neutral, and acidic solutions three times until a stable baseline was obtained. 

We then tested the MIPs-grafted 10-mer peptide chips for their affinity in order to differentiate between molecular binders of the template and their mother proteins. We observed the mass change for the HIV PR_85–94_ peptides, the His-tagged HIV PR, and the HIV PR solution at various aqueous mixtures of solutions (0.01, 0.1, and 1 ng/mL) on the HEMIPs-QCM sensor at different ratios of the aqueous mixture solution (TFE/ACN/DI water). The K_d_ value was obtained from the last two spots linear response of mass shifts and template concentrations [30,31]. They are used to evaluate the maximal capacities of HEMIPs-QCM chips as well as their binding affinities. The K_d_ value of our resultant HIV PR_85–94_ QCM chip relative to HIV PR_85–94_ itself was calculated to be 160, 1620 and 1180 pM, and B_max_ was 9.75, 7.43 and 13.2 pmole/mg for TFE:DI = 7:3, TFE:DI = 3:7, and ACN:DI = 5:5; we observed 43.3, 180 and 140 pM for His-tagged HIV PR (11.9 kDa Mw), and 26.8, 31.2 and 39.2 pmole/mg for B_max_; and we observed 78.5, 86.8 and 100 pM for HIV PR, and 35.4, 35.3 and 48.8 pmole/mg for B_max_ (Figure 3).

Interestingly, the binding affinity was highest for the HIV PR_85–94_ peptides, His-tagged HIV PR and HIV PR in the aqueous solution mixture at TFE:DI = 7:3 when compared to the solutions at TFE:DI = 3:7 or ACN: DI = 5:5. All K_d_ values were in the picomolar range, indicating an extremely strong polymer–template interaction on the chip. The results show that a high-affinity HEMIPs-QCM sensor developed for molecular discrimination when the HEMIPs film involved the aqueous solution mixture at TFE:DI = 7:3. We also noted that the binding affinity of the HIV PR_85–94_ QCM sensor for large molecular weight proteins was clearly higher than the corresponding binding affinity for small molecules. Calculated K_d_ constants for the proteins in His-tagged HIV PR and HIV PR at TFE:DI = 7:3 were 43.3 pM and 78.5 pM, respectively. Furthermore, the modified HEMIPs-QCM sensor at TFE:DI = 7:3 reached an even lower detection limit of 0.1 ng/mL for the detection of both His-tagged HIV PR and HIV PR. The results demonstrate that the specific binding and detection limits were even higher than they were for the reported immunoassay methods [32,33]. The HEMIPs’ high affinity for their mother proteins was also observable insofar as protein complex formation can occur in imprinting. It is therefore possible that the complementary complex formations emerged owing to high similarities between His-tagged HIV PR and HIV PR structures, helping fill a single epitope recognition site or cavity. HEMIPs chip captures protein at a defined orientation with minimum interference. In this regard, our assay exhibited a good binding affinity between the HIV PR and HEMIPs chip. These analytical methods rest exclusively on the specific binding affinity as antibodies and antigens to avoid too much interference from nonspecific binding. Such binding interactions are also considered as flexible epitope cavities that can adapt the native protein conformations during inclusion/exclusion in the receptor/sensor layer.

### 3.4. Specificity of HIV PR_85–94_ Templated HEMIPs-QCM Chip

We conducted a specificity analysis of HEMIPs-QCM chip toward the template peptide HIV PR_85–94_, His-tagged HIV PR, HIV PR, and three reference compounds: albumin, chymotrypsin, and papain. Table 1 illustrates the specificity of our HEMIPs chip, indicated with frequency shifts. Other analyte solutions failed to induce frequency shifts on the QCM. At concentrations lower than 1 ng/mL, the analyte solutions albumin, chymotrypsin, and papain were unable to induce frequency shifts on the QCM. Our results indicate that the helical cavities in HIV PR_85–94_-HEMIPs chip exhibited no significant differences in the adsorption amount levels of other analytes. In fact, we were able to use of our HEMIPs-QCM chips repeatedly by releasing the analytes absorbed in HEMIPs chip. The frequency shifts of the HIV PR_85–94_ templated HEMIPs-QCM chip were shown in Table 1.

### 3.5. Screening for HIV PR Inhibitor

A helical epitope peptide was selected from the α-helix structure of the HIV PR as the template molecule (Figure 4a). During polymerization, the polymer surfaces were stamped with the 10-mer peptide template. The removal of the template exposed the imprinted recognition sites that are complimentary to the template in terms of the shape, size, and distribution of functional groups (Figure 4b). Then, the special binding to HIV PR (Figure 4c) and subsequently its inhibitor (Figure 4d) in the context of HEMIPs-QCM chips were examined (Figure 4e).Then, the special binding to HIV PR (Figure 4c) and subsequently its inhibitor (Figure 4d) in the context of HEMIPs-QCM chips, and the HEMIPs chip’s use to improve drug targeting for HIV therapy to seek inhibitors for HIV PR were examined (Figure 4e). Nelfinavir was chosen as a model to undertake the HEMIPs chip, HIV PR, and HIV PR inhibitor sandwich assays. The optimized HIV PR_85–94_ HEMIPs-QCM chip operated at HIV PR adsorption. It shows that when the 1 μg/mL of HIV PR solution was injected, the frequency of the HEMIPs-QCM sensor decreased rapidly and reached a steady value within 3 min, then the recognition of the HIV PR/HEMIPs chip towards the corresponding 100 ng/mL of nelfinavir was also characterized in situ (Figure 4). 

The screening of the HIV PR inhibitor (nelfinavir) on the imprinted film was studied via the HIV PR/HEMIPs chip (Figure 5A). At first, 1 μg/mL HIV PR solutions were injected into the HEMIPs-QCM chip (arrow 1 at the blue and red lines), the frequency changes of the chips were monitored. Upon saturation, nelfinavir was injected (arrow 2 at the blue and red lines) at a concentration of 100 ng/mL until reaching equilibrium. Based on the platform of helical epitope imprinting, the complexation of the HIV PR/HEMIPs chip with nelfinavir resulted in an approximately 40 Hz or 50 Hz frequency shift, respectively. To determine the test–retest repeatability and interplatform reproducibility, an acetic acid solution was used to remove the remnants of the reactants on the HIV PR/HEMIPs chip (Figure 5A). Thus, 0.5% (arrow 3 at the blue line) or 0.8% (arrow 3 at the red line) acetic acid solution was tested to remove the nelfinavir on the HIV PR/HEMIPs chip. No nelfinavir release was observed for 0.5% acetic acid solution (blue line), while washing with 0.8% acetic acid solution (red line) released the nelfinavir from the HIV PR/HEMIPs chip until it reached the former baseline level. The results indicated that the 0.8% acetic acid (arrow 3 at red line) displayed higher extraction recoveries for nelfinavir without releasing HIV PR. The reuse of the HIV PR/HEMIPs chip can be operated for the binding and unbinding of nelfinavir in nelfinavir/HIV PR/HEMIPs assay. Therefore, after a short wash with a 0.8% acetic acid solution, bound nelfinavir was completely removed from the surface of the HIV PR/HEMIPs chip. Obvious differences in the binding baseline restoration were noted in the 0.8% acetic acid solution-treated chip when compared to the 0.5% acetic acid solution-treated chip. Reuse of the HIV PR/HEMIPs chip with 1 ng/mL or 100 ng/mL of nelfinavir was able to regenerate the “sandwich” layers for recapturing and detecting nelfinavir again (Figure 5B). High reusability performance can be attended to observe the same recognition signals. The inhibition constant K_i_ of the nelfinavir inhibitor [34] relative to HIV PR/HEMIPs chip was determined to be 1.99 nM (Figure 5C), indicating that the affinity of nelfinavir to the HIV PR/HEMIPs chip was strong enough to be used as an anchoring inhibitor. The presence of detergent can wash away the binding materials out of the background levels, without the dissociation of HIV PR from HEMIPs chip. Moreover, the concentration of nelfinavir can be determined at a clinically relevant concentration ranging from 1 ng/mL to 100 ng/mL; thus, proving this assay is suitable for screening the level of nelfinavir in clinical samples. 

### 3.6. High-Throughput Drug Discovering for HIV

A similar method can be applied to the high throughput screening for new drugs of HIV. This HIV PR/HEMIPs chip can be used to screen other compounds as potent HIV PR inhibitors. It is also feasible to screen for new inhibitors using other protein/HEMIPs chip.

## 4. Conclusions

In summary, we succeeded in fabricating a 10-mer helical epitope on QCM chips involving molecular imprinting technology. The helical epitope-templating procedure operated in optimal conditions can efficiently generate helical cavities with a high affinity for template molecules and target protein. Therefore, the HIV PR_85–94_ peptide-based HEMIPs chip successfully differentiated template molecules and target proteins from other analytes. The net binding affinities, as given by the mass change of PR bound to the HEMIPs, is proportional to the known binding constants calculated. These results suggest that a HEMIPs-based method could help orientate HIV PR on QCM surfaces and still be flexible enough to induce conformational change for binding inhibitors. Therefore, this sandwich assay system is a unique tool for detecting helical proteins/enzymes, such as HIV PR. HIV PR-attached HEMIPs-QCM chips are proven to be a super recognition element for the specific detection of nelfinavir (its inhibitor). In turn, it will be very convenient and simple to carry out a high throughput screening of other inhibitors for new drug discovery.

## Figures and Tables

**Figure 1 sensors-20-03592-f001:**
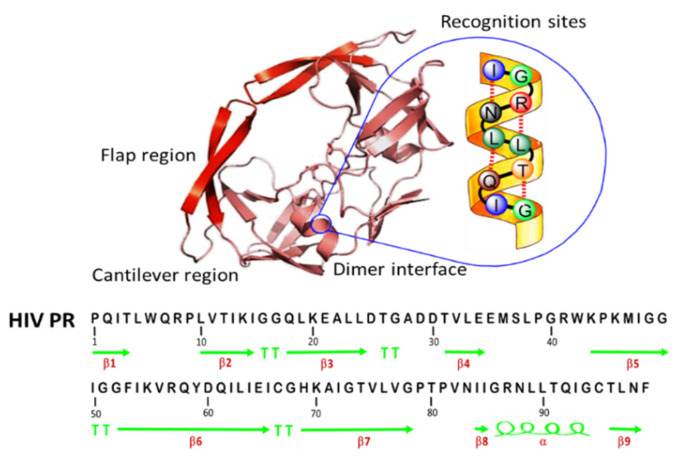
Crystal structures of HIV protease (HIV PR). HIV PR is a homodimer and contains mainly β-sheet and one α-helix per monomer. The one subunit of the monomer is shown in red and pink. The helical peptide epitope structure was selected from the α-helix structure of the HIV PR (yellow).

**Figure 2 sensors-20-03592-f002:**
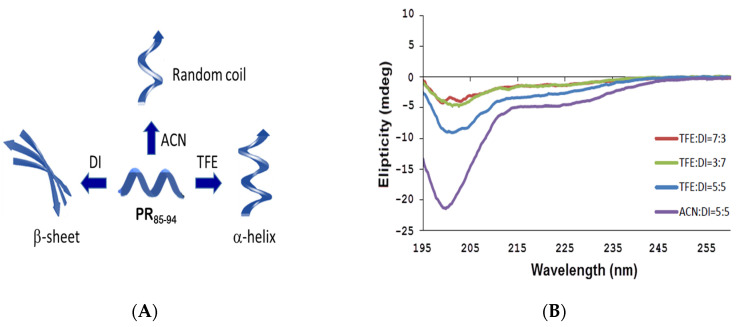
(**A**) Effects of organic solvents on protein secondary structure and function. (**B**) Circular dichroism (CD) spectra of the HIV PR_85–94_ peptides and their respective complexes on selective solvents. TFE, 2,2,2-trifluorethanol; DI, deionized water; ACN, acetonitrile.

**Figure 3 sensors-20-03592-f003:**
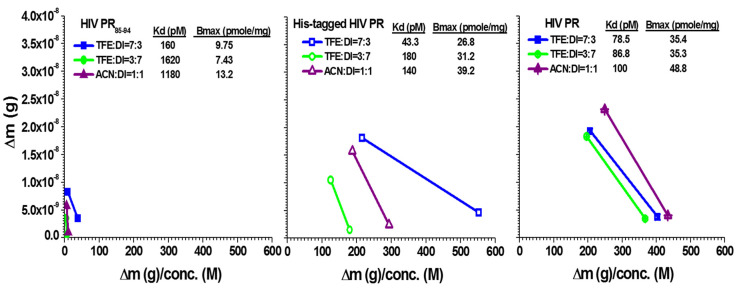
Adsorption effects of the HIV PR-HEMIPs chip. The dissociation constants and the maximum frequency shift were observed [30,31]. Each graph was observed using three selective solvents for three kinds of analytes, including template solutions: HIV PR_85–94_, His-tagged HIV PR, and HIV PR. The injected concentration of the analytes were 0.01, 0.1, and 1 ng/mL, respectively. Δm, the mass difference on the quartz surface. PR, protease; HEMIPs, helical epitope-mediated molecularly imprinted polymers. TFE, 2,2,2-trifluorethanol; DI, deionized water; ACN, acetonitrile.

**Figure 4 sensors-20-03592-f004:**
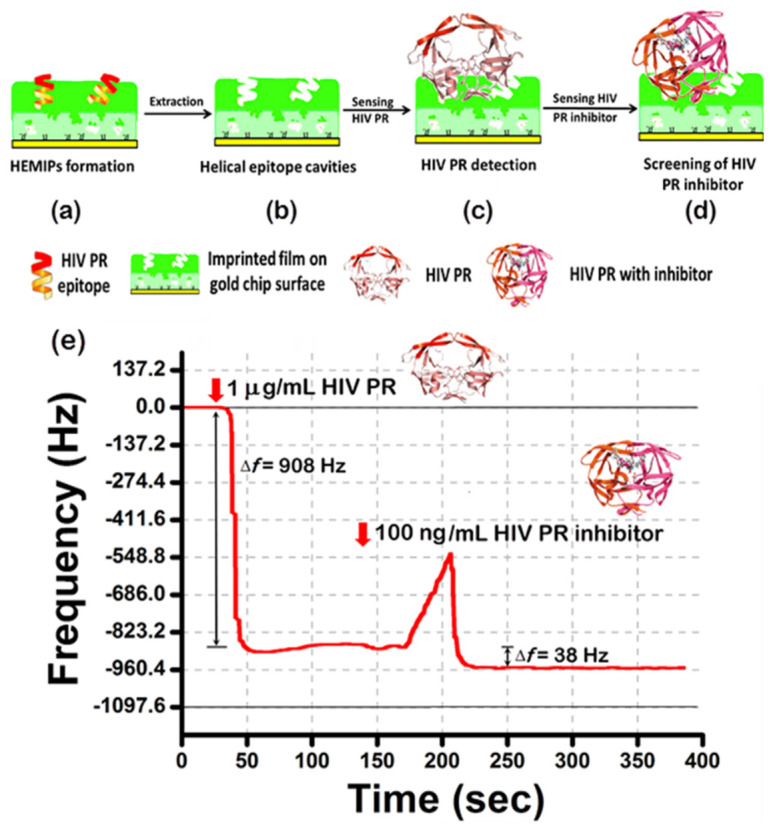
Schematic illustration of (**a**) the formation of helical epitope-mediated molecularly imprinted polymers (HEMIPs), (**b**) removal of the template molecules, (**c**) detection of HIV PR, (**d**) detection of the HIV PR inhibitor using HEMIPs on QCM chips, and (**e**) the response profile for the consecutive binding of the HIV PR and the HIV PR inhibitor to the HEMIPs-QCM chip.

**Figure 5 sensors-20-03592-f005:**
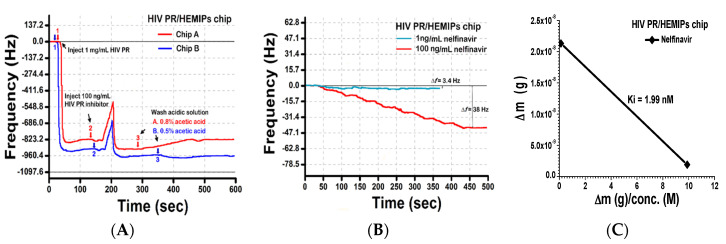
The detection of a HIV PR/HEMIPs chip for HIV PR and the screening of its inhibitor nelfinavir. (**A**) Frequency shifts of injecting HIV PR and its inhibitor nelfinavir solution into a HIV PR-HEMIPs chip, followed by washing with different concentrations of acetic acid solution as detergent. (**B**) Reuse of the HIV PR/HEMIPs chip towards 100 μL of 1 ng/mL and 100 ng/mL nelfinavir after washing with 0.8% acetic acid solution. (**C**) Ki value for the binding of the inhibitor to the HIV PR. Δm, the mass difference on the quartz surface.

**Table 1 sensors-20-03592-t001:** Specificity of the HIV PR_85–94_ templated HEMIPs-quartz crystal microbalance (QCM) chip.

Analyte			HEMIPs-QCM Chip (Hz)
HIV PR_85–94_	0.1 ng/mL		6.3
		1 ng/mL	14.8
His-tagged HIV PR	0.1 ng/mL		8.3
		1 ng/mL	32.4
HIV PR	0.1 ng/mL		6.7
		1 ng/mL	34.3
Albumin	10^6^ ng/mL		---
Chymotrypsin	10^6^ ng/mL		---
Papain	10^6^ ng/mL		---

Abbreviations: PR, protease; HEMIPs, helical epitope-mediated molecularly imprinted polymers; QCM, quartz crystal microbalance. ---, no frequency shift was observed.

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
