# Peer review of "Sensing HIV Protease and Its Inhibitor Using “Helical Epitope”—Imprinted Polymers"

_sensors, 2020, doi:10.3390/s20123592_

Round 1
Reviewer 1 Report
The article details the manufacture of MIP with a very good detection limit and promising results in the search for HIV-PR inhibitors. However, there are important details that must be clarified before publication.
First, the authors should include a scheme of chip manufacture as it is not understood what they use the (N-Acr-L-Cys-NHBn)2 solution for. Furthermore, it is not explained whether the orientation of the epitope influences detection.
On the other hand, small peptides or other peptides with similar structure that could influence are not evaluated in cross-reactivity studies. Similarly, it is not studied whether other substances could influence the determination of protease inhibitors even though they did not bind to the protease.
That is why I think that, mainly this last comment, needs to be clarified before publication.
Author Response
We greatly appreciate the reviewer #1 comments, which gave us the opportunity to improve the manuscript. We believe that the clarity and the quality of this revision has been significantly improved in the revised version. Point-by-point responses are listed following every specific comment in this response letter. The revised parts of the manuscript (our responses) are represented in red, bold, and underline in the text.

Reviewer 2 Report
In this manuscript authors report the development of a helical epitope molecularly imprinted polymers (HEMIPs) integrated with a quartz-crystal microbalance (QCM) sensor transducer using a helical epitope-peptide (peptide HIV PR85-94) isolated from the helix structure of the HIV protease as the template molecule. The HEMIPs-QCM is expected to detect and quantify both target epitope-peptide as well as to an inhibitor of HIV PR (nelfinavir).
The findings are quite interesting, and results are clearly presented. The presented analytical approach could be further developed for the demanded therapy of HIV infections. Also, the manuscript is within the scope of sensors and therefore may be considered for publication after some corrections as highlighted below:
- Line 163-164; Why were those monomers selected? Was any theoretical calculation made?
- Line 175-176- For the sake of non-expert readers, the authors should establish the relationship between kd and affinity. This will enhance better comprehension in the subsequent paragraphs.
- How is the limit of detection calculated and at what concentration is the target present in real media?
- Lines 193-195; What is the reason behind such differences in the binding affinity of small and large molecules?
- Line 213 – 214; How many times can the HEMIPS-QCM chips be repeatedly used and what is the difference in binding response from one repeat to another?
- Figure number should be added to the MIP formation scheme on page 7 and should be properly referred in the manuscript text.
- Line 230-231; Figure number of frequency- time graph is wrong.
- Line 232-233; The statement and the associated fig. 5A is a little confusing in terms of its intention on two grounds:
40 or 50 Hz frequency shift is not visible from fig. 5A. After injecting nelfinavir, almost 300 Hz shift is observed.
b. what does the author mean by "respectively" in this sentence? - The experimental facts and explanations to Fig. 5A need better organization;
g.
a. why is there an upward shift instead of a frequency decay following inhibitor injection?
b. the explanation given in line 235-238 is not clear. From Fig 5A, it seems both 0.5 and 0.8 acetic acid solution lead to nelfinavir removal as it appears both signals return to baseline.
Author Response
We greatly appreciate the reviewer #2 comments, which gave us the opportunity to improve the manuscript. We believe that the clarity and the quality of this revision has been significantly improved in the revised version. Point-by-point responses are listed following every specific comment in this response letter. The revised parts of the manuscript (our responses) are represented in red, bold, and underline in the text.

Round 2
Reviewer 1 Report
The authors have made the necessary changes and answered the questions raised by the reviewer. Despite the fact that in the text the cross-reactivity could be a little better explained, the authors have correctly justified their decision. Therefore, I consider that the article can be accepted.